# Modeling the Physical Properties of Gamma Alumina Catalyst Carrier Based on an Artificial Neural Network

**DOI:** 10.3390/ma12111752

**Published:** 2019-05-29

**Authors:** Hasan Sh. Majdi, Amir N. Saud, Safaa N. Saud

**Affiliations:** 1Department of Biomedical Engineering, Al-Mustaqbal University Collage, Babylon, Iraq; hasanshker1@gmail.com; 2Faculty of Information Sciences and Engineering, Management & Science University, Shah Alam 40100, Selangor, Malaysia; safaaengineer@gmail.com

**Keywords:** γ-alumina, catalyst carrier, gel-casting, artificial neural network

## Abstract

Porous γ-alumina is widely used as a catalyst carrier due to its chemical properties. These properties are strongly correlated with the physical properties of the material, such as porosity, density, shrinkage, and surface area. This study presents a technique that is less time consuming than other techniques to predict the values of the above-mentioned physical properties of porous γ-alumina via an artificial neural network (ANN) numerical model. The experimental data that was implemented was determined based on 30 samples that varied in terms of sintering temperature, yeast concentration, and socking time. Of the 30 experimental samples, 25 samples were used for training purposes, while the other five samples were used for the execution of the experimental procedure. The results showed that the prediction and experimental data were in good agreement, and it was concluded that the proposed model is proficient at providing high accuracy estimation data derived from any complex analytical equation.

## 1. Introduction

A catalyst is a compound capable of accelerating a chemical reaction without appearing in the final products [1]. Tiny metal particles are universally used to make multiphase catalysts, and porous ceramics are usually used as the carriers of these metal particles [2]. Therefore, the interconnected pores of these ceramics need to have a size range of 6 nm to 500 μm [3]. Alumina is the material of choice for these carriers, and titania, zirconia, silica, and silicon carbide are also used [2]. Catalyst porous ceramics can be used in methane steam reforming, oxidation of ammonia, decomposition of organics by photocatalysis, and destruction of volatile organic compounds (VOCs) by incineration. The tortuous channels in porous ceramics can generate turbulence and thus ensure good mixing of the reactants and radial dispersion [2,3]. Due to the presence of the interconnected and large pores, the accumulation of dust cannot block the pores. Compared to a reactor with filled stacking particles, a reactor with a ceramic foam core can reduce the drop in pressure. Porous ceramics also have good absorbability. The conversion and reaction rate increase significantly for the reactive fluid that flows through the porous ceramic channels [4]. Due to the thermal shock and chemical corrosion resistance of porous ceramics, they can be used in high-demand service conditions, such as automotive exhaust treatment and reactors in chemical engineering. The popularization of the use of inorganic separation membranes with porous ceramics may bring about a breakthrough in production in the chemical industry. Automobile emissions are one of the major sources of environmental pollution [5]. Porous ceramic catalysts are also widely used for exhaust purifiers due to their high specific surface area, high thermal stability and wear resistance, nontoxicity, and low density. With the application of this kind of purifier in exhaust pipes, more than 90% of the harmful CO, HC_x_, and NO_x_ in the exhaust can be converted into non-toxic CO_2_, H_2_O, and N_2_. If applied in a diesel engine, the purification rate of the carbon granules can exceed 50% [2,3]. When the ceramic foam has become full of carbon granules, they can be removed by catalytic oxidation or combustion so that the foam can be recycled [6]. The texture of the carrier must allow good dispersion of the active phase and ensure the diffusion of the reagents and products between the reaction medium (liquid or gaseous) and active sites [7]. The specific surface area of a particle is a function of porosity, pore size distribution, shape, size, and roughness. The role of the specific surface area is critical in the design of a catalyst where, typically, a domain with a high specific surface area carrier (e.g., γ-alumina, silica, zeolites) is included in the structure of the catalyst. Catalyst performance is therefore determined by the specific surface area of the carrier [8].

The artificial neural network (ANN) is a newly developed and effective technique that can be used for impartial predictions [9,10]. Neural networks (NN) are parallel information-processing systems consisting of several simple neurons (also called nodes or units) that are organized in layers and connected by links. The artificial neural networks imitate the highly interconnected structures of the brain and the nervous system of animals and humans, whereby the neurons correspond to the cell bodies and the links are equivalent to the axons [11]. There are a number of different types of ANN: The Feedback ANN, in which the output goes back into the network to achieve the best results internally, the Feed Forward ANN, with a simple neural network consisting of an input layer, an output layer and one or more layers of neurons, and the Recurrent ANN, which has both a feed forward path and a feedback path in the reverse direction [11]. The artificial neuron consists of the following basic components [12,13]:A group of inputs: (x1, x2, x3…Xn), each of which has a weighting related to it;A summation function to determine the summation of the weighted input and bias;An activation function.

The summation of the weighted input is then fed through a transfer function to produce a neuron output. In this study we used the sigmoidal function, because it is a nonlinear function and can be expressed as follows:(1)$$y=\frac{1}{1+e^{-x}}$$

Numerous studies have been conducted on neural networks as prediction alternatives for metal-based properties; however, for ceramic-based carriers, the implementation of ANNs is not yet ordinarily used. In the present work, an ANN numerical model is used to predict the physical properties of a porous alumina ceramic carrier [11,14]. This involves the concentration of yeast, the temperature of sintering, and soaking times as input parameters with three diverse readings taken to enhance the accuracy of the results and reduce the estimation error.

## 2. Materials and Methods

Alum (NH_4_Al (SO_4_)_2_.12H_2_O) was used as the raw material for preparing alumina. Yeast cells were the pore-forming agent, and agar was used to form the gel. 

A quantity of 0.2 M of NH_4_Al (SO_4_)_2_.12H_2_O was dissolved in 100 mL of distilled water at a temperature of 70 °C and treated with ultra-sonication for 2 h until fully dissolved. Then, the white precipitate filtration product powder was dried at 150 °C and calcined at 800 °C to obtain γ-alumina.

The porous alumina sample was prepared by the gel casting method using yeast as a pore-forming agent and agar to form a gel. First, mixtures of alumina and different ratios of yeast cells (0, 2, 10, 20 wt.%) were prepared by ball milling. To prepare the gel-casting slurries, the agar solution was prepared by dissolving the agar in distilled water (the agar to water ratio was found to be 0.4 wt.%.) with the help of microwave treatment for 1 min using 900 W of power and a frequency of 2450 MHz. The solution was then transferred to a water bath at 55 °C. The desired amount of mixture (alumina with yeast) was added to the agar solution at 55 °C under sonication and mixed at a speed of 1600 rpm for 15 min. After that, the slurry was poured in PVC molds and then cooled to 10 °C in the refrigerator. The alumina gel was then demolded and aged at room temperature for 24 h and dried in a convection oven at 55 °C. Samples were then sintered at 500 °C and 900 °C. Four different batches with different concentrations of yeast, sintering temperatures, and socking times were used to prepare the porous alumina specimens. The surface area was measured by a surface area and pore analyzer (Thermo Scientific, Waltham, MA, USA) which was available in the Universiti Teknologi Malaysia, Johor Bahru, Malaysia for pores in the range of 20 Å to below ~1500 Å [15].

Matlab (trial version) software (MATLAB R2019a, Mathworks, Natick, MA, USA) applications were used in this work to simulate the behavior of artificial or biological neural networks.

## 3. Physical Properties

### 3.1. Apparent Porosity

The specimens were tested by the Archimedes method according to American Society for Testing and Materials (ASTM C373-88). The porosity is the ratio of open pores to exterior volume and can be determined as follows [16]:(2)$$P=\left(\frac{M-D}{M-S}\right)*100\%.$$

### 3.2. Bulk Density

The bulk density of specimens was calculated after sintering according to ASTM C373-88 by the following equation [16]:(3)$$\rho =\left(\frac{D}{V}\right)$$

### 3.3. Sintering Shrinkage

The shrinkage that occurs in the specimen as a result of the sintering process can be determined by measuring the outer diameter before and after sintering with a caliper and then using following equation:(4)$$\mathrm{Shrinkage}\;=\left(\frac{D1-D2}{D1}\right)*100\%$$
where V is the exterior volume (cm^3^), M is the saturated water weight (g), D is the dry weight (g), S is the suspended weight (g), D1 is the dimension of the specimen before shrinkage (mm), D2 is the dimension of the specimen after shrinkage (mm), P is the apparent porosity (%), and ρ is the bulk density (g/cm³).

## 4. Artificial Neural Network (ANN) Model

To cultivate a predictive model, it was essential to arbitrarily distribute the accessible data collected into training data and testing data. The data collection consisted of 30 ceramic samples. The input data took into consideration the concentration of yeast, the sintering temperature, and the soaking time, within three values, and the estimated parameters were porosity, shrinkage, density, and surface area, as shown in Figure 1. The training function used was TRAINGDX, a network training method that is updated for weight and bias values according to gradient descent momentum and an adaptive learning rate [17]. The architecture of the ANN is summarized in Table 1.

There were 25 samples implemented in the training model, and five samples implemented in the testing procedure. The data collection used for training the ANN is given in Table 2.

## 5. Validation of an Artificial Neural Network (ANN) Model

Validation of the ANN model was carried out to verify that the generalized output data contained the minimum mean error when compared to the experimental data. Once the model had completed the training process successfully, it was ready to predict the essential data. Table 3 summarizes the validation of the model for porosity, density, shrinkage, and surface area. It was found that the validation results were achieved with few relative mean errors, which verified that the prediction and experimental data were in good agreement. According to the model architecture, the input data of porosity and density were related to each other with the same trend. Thus, the output results confirmed that the ANN model is a tool with a high potential for predicting the physical properties of porous alumina. Figure 2 shows the relationship between the correlation coefficient R and the independent variables, which match well with the experimental data. The dashed line in Figure 2 refers to the perfection of the predicted results and experimental data, whereas the circles and lines represent the data points and best-matched results, respectively. It was also revealed that the error gap between the dashed line and experimental data is quite small, which supports the accuracy of the attained results. The following guide was suggested to verify the model correlation coefficient results between experimental and predicted values [18]: When R < 0.8, there is a strong correlation; when 0.2 < R < 0.8, a correlation exists; and when R ≤ 0.2, there is a weak correlation.

## 6. Results and Discussion

### 6.1. The Effect of Sintering Temperature

Figure 3 shows the strong correlation between the sintering temperature and porosity, density, shrinkage, and surface area. It was shown that an increase in the sintering temperature led to an increase in the convergence between the grains and a reduction in the pore size. This produced a denser ceramic structure and consequently, the porosity decreased. Caiyun Jia et al. [19] also showed that as the sintering temperature increased from 900 to 1600 °C, the porosity of the foam alumina decreased from 82.47% to 74.92%, due to an increase in the grain size. This permitted the grains to be merged and assembled, thus reducing the open porosity. Furthermore, Wu Qin et al. [20] also revealed that raising the sintering temperature from 1450 to 1600 °C led to a decrease in the open porosity from 55.35% to 49.08% as a consequence of the sintering shrinkage and grain growth, which may result in narrowing the gap between the alumina particles. However, the shrinkage and density of the alumina tended to increase as the sintering temperature increased, and this is mainly attributed to a decrease in the pore size of the sintered body [13].

### 6.2. The Effect of the Yeast Concentration

Figure 4 shows the correlation between the concentration of yeast and the porosity, density, shrinkage, and surface area. It was shown that an increase in the concentration of yeast led to an increase in the ratio of volatility of yeast cells. This led to increases in the porosity and surface area. However, the density and shrinkage decreased. Guogang Xu et al. [21] also showed that raising the ratio of yeast to alumina from 1:2 to 1:1% increased the percentage of open porosity and produced large pore sizes of up to 3 μm.

### 6.3. The Effect of Soaking Time

As shown in Figure 5, the predicted density substantially increased as the soaking time increased. This is because the increase in the soaking time will make the grains converge more and increase their contact with each other. This reduces the porosity and surface area of the final sintering porous structure. K. Darcovich [17], also showed the effect of soaking time on porosity; as the soaking time increased, the densification of the bodies increased. For the samples prepared at lower temperatures, however, the porosity showed less of a dependence on soaking time.

## 7. Conclusions

A model of an artificial neural network for prediction of the properties (porosity, density, shrinkage, and surface area) of porous γ-alumina prepared by the gel-casting method was designed. The input parameters of the model were the sintering temperature, the concentration of yeast cells, and the soaking time, while the outputs were porosity, density, shrinkage, and surface area. Considerable savings in cost and time were achieved using an artificial neural network (ANN) model. The artificial neural network results showed convincing agreement with the experimental data and the ANN provided further useful data. The log sigmoid activation function was proficient at predicting the values of physical properties.

## Figures and Tables

**Figure 1 materials-12-01752-f001:**
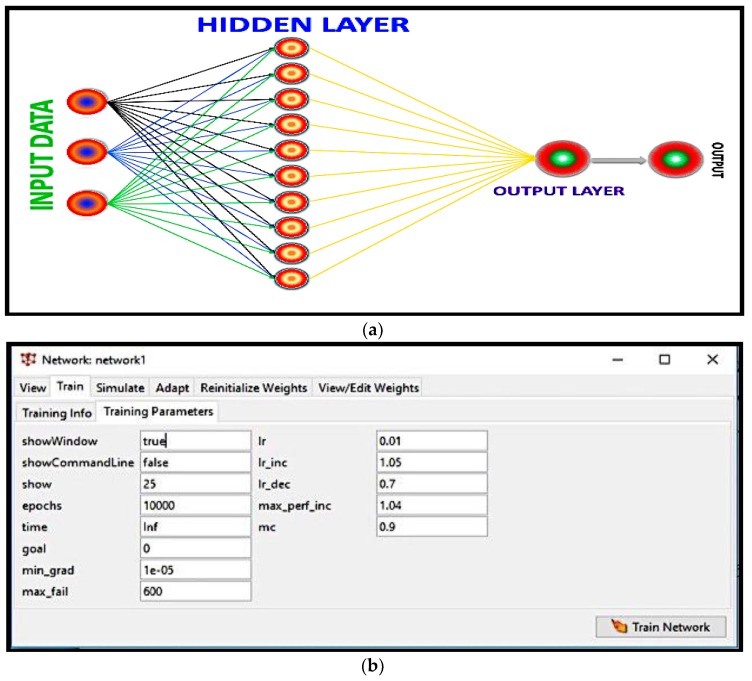
ANN model (**a**) architecture view and (**b**) training parameters.

**Figure 2 materials-12-01752-f002:**
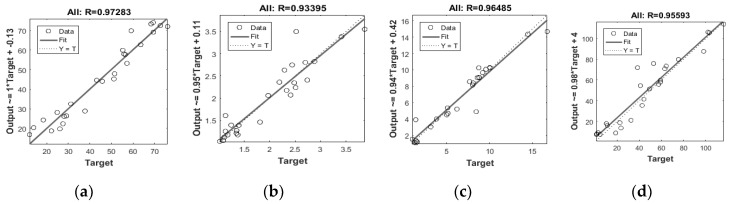
Neural network training regression for (**a**) porosity, (**b**) density, (**c**) shrinkage, and (**d**) surface area.

**Figure 3 materials-12-01752-f003:**
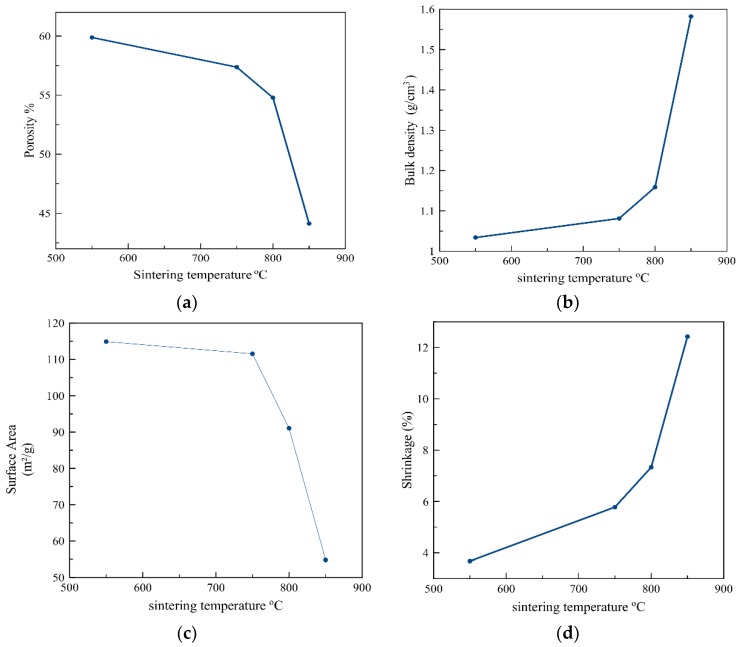
Simulated effect of the sintering temperature on: (**a**) Porosity, (**b**) Bulk density, (**c**) Surface area, and (**d**) Shrinkage.

**Figure 4 materials-12-01752-f004:**
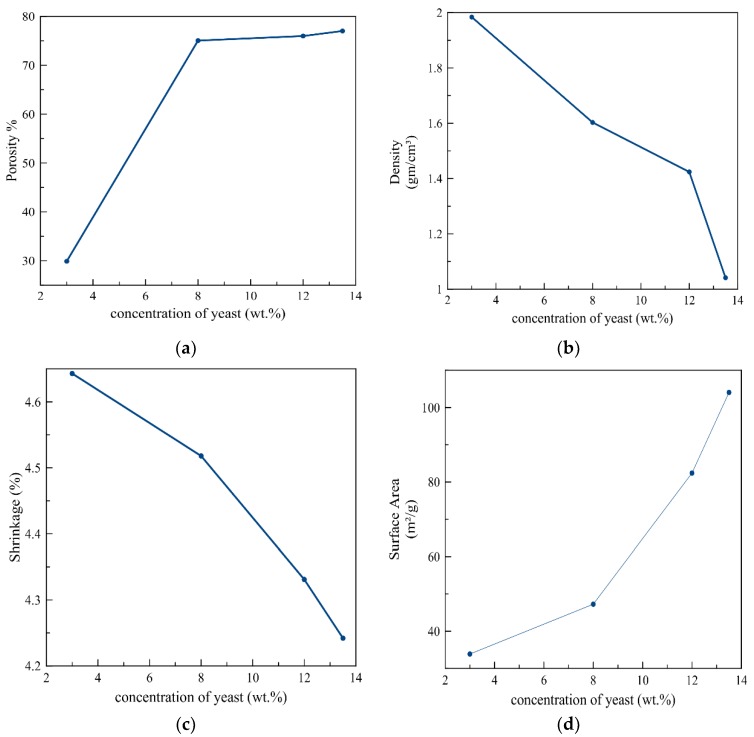
Simulated effect of the concentration of yeast on: (**a**) Porosity, (**b**) density, (**c**) Shrinkage, and (**d**) Surface area.

**Figure 5 materials-12-01752-f005:**
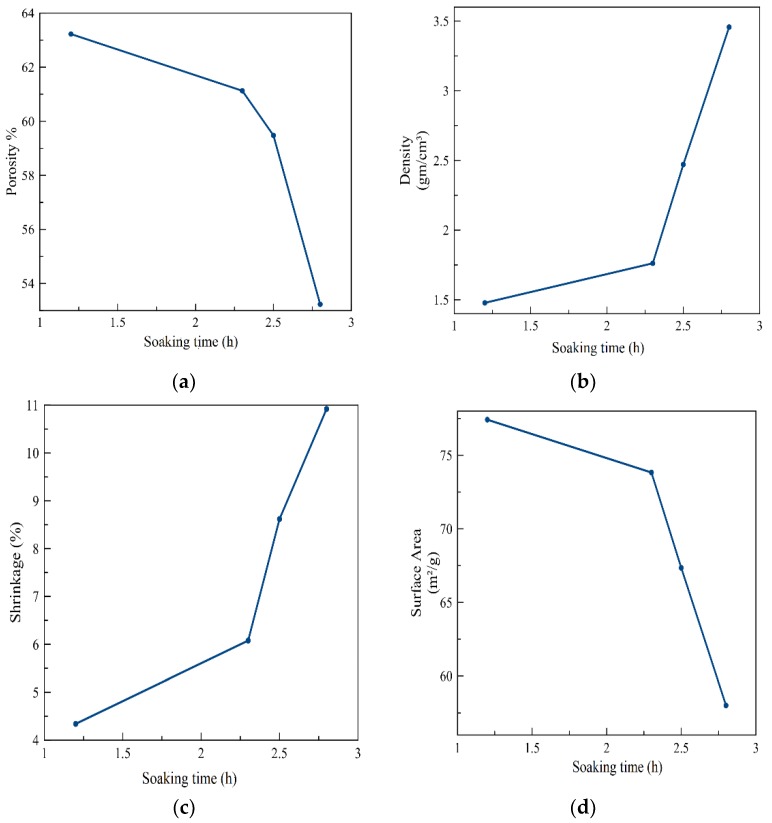
Simulated effect of the soaking time on: (**a**) Porosity, (**b**) density, (**c**) Shrinkage, and (**d**) Surface area.

**Table 1 materials-12-01752-t001:** ANN parameters for the indicated model.

ANN Parameter	Values
Network type	Feed Forward Back Propagation (FFBP)
Training function	TRAINGDX
The number of layers	Two layers
The number of the nodes in each layer	input: 3, hidden: 10, output: 1
Activation functions	Log sigmoid
The initial weights and biases	A random value between −1 and +1

**Table 2 materials-12-01752-t002:** The processing parameters of the training data.

The Concentration of Yeast wt.%	Sintering Temperature (°C)	Socking Time (h)	Porosity %	Density (g/cm³)	Shrinkage (%)	Surface Area (m²/g)
2	500	1.5	31.2	2.5	8.5	32.6
2	500	3.0	28.1	2.4	8.8	23.6
2	700	1.5	29.2	2.3	8.5	22.5
2	700	2.0	27.6	2.4	9.2	18.8
2	700	3.0	24.9	2.5	8.9	11.2
2	900	1.5	26.1	2.2	8.8	10.4
2	900	2.0	22.2	2.7	10.0	4.7
10	500	1.5	57.2	1.3	3.9	63.2
10	500	2.0	55.8	1.4	5.2	59.3
10	500	3.0	51.4	2.0	8.2	52.9
10	700	1.5	56.4	1.3	5.0	59.3
10	700	3.0	45.8	2.4	7.8	38.5
10	900	1.5	51.1	1.8	6.3	41.0
10	900	2.0	43.2	2.5	8.0	49.4
10	900	3.0	37.8	2.7	9.6	42.7
20	500	1.5	75.9	1.0	1.7	116.3
20	500	2.0	69.4	1.1	1.4	101.9
20	500	3.0	69.3	1.1	1.5	98.2
20	700	1.5	72.7	1.0	1.0	103.3
20	700	2.0	68.3	1.2	1.7	75.5
20	900	1.5	63.6	1.1	1.4	64.3
20	900	2.0	59.1	1.2	3.2	57.5
20	900	3.0	55.2	1.4	5.1	44.3
0	500	1.5	18.5	2.9	10.0	2.9
0	700	1.5	13.9	3.4	14.5	1.8
0	900	1.5	12.1	3.8	16.7	1.3

**Table 3 materials-12-01752-t003:** Validations of the ANN model.

Porosity (%)			Density (g/cm^3^)			Shrinkage (%)			Surface Area (m²/g)		
Pre.	Exp.	Relative Error	Pre.	Exp.	Relative Error	Pre.	Exp.	Relative Error	Pre.	Exp.	Relative Error
23.3	25.1	−0.07	1.7	2.2	−0.29	9.3	8.7	0.06	28.1	29.4	−0.04
49.2	49.3	−0.002	1.9	2.3	−0.21	6.4	5.7	0.10	43.2	38.7	0.10
63.9	61.7	0.03	1.2	1.1	0.08	1.7	2.0	−0.17	59.3	62.1	−0.04
16.0	15.1	0.05	3.5	3.9	−0.11	14.9	16.7	−0.12	7.5	6.3	0.16
22.3	19.8	0.10	3.6	2.9	0.19	11.8	9.9	0.16	3.4	3.2	0.05

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
