# Peer review of "Modeling the Physical Properties of Gamma Alumina Catalyst Carrier Based on an Artificial Neural Network"

_materials, 2019, doi:10.3390/ma12111752_

Round 1
Reviewer 1 Report
The work is relevant and has great prospects for development. This manuscript describes a new approach in the development of alumina supports — the use of neural networks to predict their physicochemical properties and parameters of the porous system. Self-learning neural networks are successfully used for modeling of biological objects. The application of these methods in the field of inorganic chemistry and materials science will be of great interest among researchers.
Despite all the advantages of the article, it should be carefully revised and supplemented. In the current form, the manuscript CAN NOT be accepted for publication.
I have some remarks and suggestions:
1. The introduction must be corrected. The sentences in the introduction, revealing the catalyst and dividing it by function, in my opinion, are superfluous. It is necessary to note the relevance of the hydrogenation processing of petroleum fractions, including high-boiling fractions and oil residues. It is necessary to develop the idea of requirements for increased catalyst porosity, which result from processing of heavy oil fractions.
2. Not described the method of the support preparation, and also what initial chemicals and the materials were used. The characteristics of the supports studied are not given also:
- difractogram (as confirmation phase composition, which is specified in the title of the article);
- are these granules (then what shape and size) or microspherical powder (dimension). Also it would be good to bring micrographs of the support (SEM).
3. Not given the methods and equipment for the study of alumina supports: to determine the porosity, shrinkage, density. What kind of density was determined (bulk, true or apparent).
4. In the manuscript there is no description of the neural network, its type, principles, mathematical apparatus. What kind of computer program was used? It is necessary to explain the choice of a particular neural network.
5. The explanation of the observed physical phenomena must be confirmed by the results of experiments, not only by references to the works. For example, when it comes to sintering of the alumina as a result of the closure of pores and the growth of primary crystallites of alumina (not just “grains”), it is necessary to give the sizes of these primary crystallites, calculated on the basis of diffractograms, and also to give the pore size distribution.
Inaccuracies and typographical errors
The presentation of material requires a significant improvement, there are many inaccuracies, misprints, incorrect sentences.
1. The correlation coefficient is given in% and fractions. Need to choose only one.
2. What is the error of methods for determining the specific surface, porosity, shrinkage? How important are the tenths and hundredths of a percent for the values of these indicators? Based on this, in the table it is necessary to round the values to tenths or whole values.
3. It is necessary to make a separate graph for the density or use separate Y axis, since on the current graph it is not clearly visible that the density increases 1.5 times.
4. Check the numbering of pictures
5. The title of Table 1is missing.
English should be greatly improved. There are incorrectly translated words, for example, “inexpressive”, “predicate“, “predication” and incorrect expressions, for example, “effective developed technique”, “grains”.
The sentences below are not finished:
Lines 103-105: “However, the shrinkage and density of the alumina tend to increase as the sintering temperature increased and this is mainly attributed”
Lines 114-115: “However, the shrinkage and density of the alumina tend to decrease as the ratio of yeast to alumina increased and this is mainly attributed”
Author Response
I would like to thank the reviewer for the comments provided which reinforce the diligence of the paper, and where all the points provided by the reviewer were corrected
I would like to point out to the reviewer that the idea of the research is to build a mathematical model using the neural network in the Matlab program, based on the results of the pre-recorded process that referenced [13]
The corrections and answers to the questions were listed in the following attached file
Thank you so much

Reviewer 2 Report
The authors describes the employ of a predictive model divided in training data and testing data applied to artificial neuronal network.
In the introduction a major emphasis it has been given to describe alumina as catalyst. Follow a minor description of the topic of interest as artificial neural network. For this reason is indicate to reformulate the introduction with less sentence and describe properly the predictive model involve with references as example and applicative use, if not yet use with biologicals example
In line 54. 3. Artificial neural network. It is not reported an esaustive description of this model and the apply of data collection is affected by a do not understand of how this model is apply.
In Figure 1 is indicate to correct the figure with a properly a) and b)
In Figure 1. it is not clear this representation in particular in b) there are some parameters of a program, but no specific indications was add.
Author Response
I would like to thank the reviewer for the comments provided which reinforce the diligence of the paper, and where all the points provided by the reviewer were corrected
The corrections and answers to the questions were listed in the following attached file
Thank you so much

Round 2
Reviewer 1 Report
Thank you for your answers. The manuscript needs significant revision. I want to help you to make the manuscript better, so that the presentation of the material will correspond to the high level of the journal “Catalysts”.
My comments:
1) This article is an independent work, therefore it is necessary to indicate in the experimental part all the methods that were used to obtain the characteristics of the support, which used in the calculations.
One more note, in the reference [13] there are no names and types of the equipment on which the following characteristics were obtained are not indicated: specific surface area and pore volume, shrinkage, density. Also not indicated errors in the definition of these characteristics. What kind of the density you determined and used for calculations in your manuscript?
2) What does the "porosity" mean? How you determined this parameter? Did you use the pore volume, determined from the nitrogen adsorption-desorption? If yes, in this case you considered only pores smaller than 150 nm. If no, how you determined the pores larger 150 nm (as I can see from the ref. [13] this macropores contribute greatly to porosity, perhaps the main).
3) Since your main task is to use an artificial neural network in calculating the characteristics of inorganic porous materials, it is necessary to describe in more detail the ANN used by you.
What is the mathematical apparatus of the artificial neural network. Add please it to the experimental. The software, which you used for calculations also should be added to the experimental section (uncluding the version of the software). In the current version of the manuscript you only indicated in the introduction that the matlab can be used for calculations. in my opinion this phrase in the introduction is inappropriate. If you don’t have the license of the software, you should to define that you used the trial version.
What do the letters FFBP and TRAINGDX mean? As well as terms terms “Log sigmoid activation function”, “biases”? for non-specialists in the field of artificial neural networks, these terms and abbreviations will not be clear. It is necessary to give a brief transcript and description (perhaps mathematical).
4) you use two terms – “support” and “carrier”. Choose one.
5) lines 27-29: “For example, the selection rate of the oxidative products for the ethylene increases from 65% to 80% with the alumina carrier in the Ag/a-Al2O3 system [2]” - expression is not consistent
6) lines 38-39: “…like the auto exhaust treatment and the reactor in chemical engineering.” - expression is not consistent.
7) line 45: use subscript.
8) line 59: “There are a number of different types of ANN : Feedback ANN, Feed Forward ANN ,and Recurrent ANN [11]”. Add one-two sentences, what is the difference between them.
9) What is the function of Figure 1b, what does it explain?
10) I found the link for the Journal of Engineering and Applied Sciences (from ref. [13]).
http://medwelljournals.com/archive.php?jid=1816-949x
This journal doesn’t has the vol. 18 for 2018, as indicated in your manuscript. May be I found the wrong journal. So, can you provide me please the link to the journal from the reference [13] with archives of volumes.
One more note, this reference doesn’t has DOI, so for some readers it could be complicated to find this reference.
11) There are inconsistencies in the numbering of sections.
12) Lines 119-120: “However, the shrinkage and density of the alumina tend to increase as the sintering temperature increased and this is mainly attributed to a decrease in the pore size of the sintered body”.
This assumption is controversial, because usually under sintering the pore size increase and the reason of the sintering is the diffusion of the Al atoms and reorganization of the crystal structure of aluminum oxide into a more close-packed. You should to provide the references of your assumption.
13) “All data are placed in a single graph to show the effect of each factor on all the results at once”.
In this case we get a distorted view. For example as we can conclude from the Fig. 3, the porosity decreases 1.4 times. At the same time, shrinkage increases 3.4 times. But due to the fact that the absolute values of porosity are greater, the changes in the figure are perceived to be comparable for porosity and shrinkage.
14) There are numerous typos, incorrect translation and unsuccessful expressions (see pdf-file, highlighted with yellow marker).
15) Figures 3 and 4 - add units of on the x-axis.
16) Table 3 is missing.
17) Conclusions “A model of artificial neural network for the physical properties’ prediction…” – Porosity and surface area are not physical properties.
18) References – they should be made according to the requirements of the journal.
Author Response
IDear reviewer
] would like to extend my thanks and gratitude to you for all the corrections that have been received and which without a doubt will make the paper stronger and better for the readers. All questions have been answered and all corrections have been made.
Many thanks for you
Amir N.Saud

Reviewer 2 Report
The manuscript is now ready to be publish in the present form. It is indicate to reserve attention to the formulas of chemical compounds.
Author Response
I would like to extend my thanks and gratitude to you for all the corrections that have been received and which without a doubt will make the paper stronger and better for the readers. All questions have been answered and all corrections have been made.
thank you
Amir N.Saud

Round 3
Reviewer 1 Report
Hello, dear authors. Thanks for making changes to the manuscript. Unfortunately, it still contains many errors. I attach a pdf-file with my comments.
Reviewing your manuscript takes a lot of time, so if next time it contains the same large number of errors (including typing errors, design errors – mainly about tables and References), then I will have to recommend the article to be rejected.
I hope you understand my desire to optimally distribute my working time.
I wish you all the best.

Author Response
I apologize very much for not making any corrections required in the second version but this was because the file is missing as shown in the picture in the attached file .
Now all the required corrections have been made which will certainly increase the strength and consistency of the research.

Round 4
Reviewer 1 Report
Dear authors, I send you again the pdf-file. Please check your English in all the text of the manuscript. And check again the references, because surnames in some references written in capital letters.
Best regards

Author Response
Dear Editor All required corrections have been modified in the attached file and in red color. Thank you so much for all the comments
